

# The origin of the period-$2T/7$ quasi-breathing in disk-shaped Gross-Pitaevskii breathers

Jordi Torrents[1], Vanja Dunjko[2], Marina Gonchenko[3],
Gregory E. Astrakharchik[4] and Maxim Olshanii[2*]

**1** Departament de Física de la Matèria Condensada, Universitat de Barcelona,
Martí i Franquès 1, 08028 Barcelona, Spain
**2** Department of Physics, University of Massachusetts Boston,
Boston Massachusetts 02125, USA
**3** Universitat de Barcelona, Barcelona, Spain
**4** Departament de Física, Universitat Politècnica de Catalunya, E08034 Barcelona, Spain

* maxim.olchanyi@umb.edu

## Abstract

We address the origins of the quasi-periodic breathing observed in [Phys. Rev. X vol. 9, 021035 (2019)] in disk-shaped harmonically trapped two-dimensional Bose condensates, where the quasi-period $T_{\text{quasi-breathing}} \sim 2T/7$ and $T$ is the period of the harmonic trap. We show that, due to an unexplained coincidence, the first instance of the collapse of the hydrodynamic description, at $t^* = \arctan(\sqrt{2})/(2\pi)T \approx T/7$, emerges as a 'skillful impostor' of the quasi-breathing half-period $T_{\text{quasi-breathing}}/2$. At the time $t^*$, the velocity field almost vanishes, supporting the requisite time-reversal invariance. We find that this phenomenon persists for scale-invariant gases in all spatial dimensions, being exact in one dimension and, likely, approximate in all others. In $d$ dimensions, the quasi-breathing half-period assumes the form $T_{\text{quasi-breathing}}/2 \equiv t^* = \arctan(\sqrt{d})/(2\pi)T$. Remaining unresolved is the origin of the period-$2T$ breathing, reported in the same experiment.



# 1  Introduction

## 1.1  The phenomenon

A quasi-periodic breathing of quasi-period $2T/7$ was observed, in experiments by J. Dalibard's group, in disk-shaped harmonically trapped two-dimensional (2D) Bose condensates [1]. Here $T = 2\pi/\omega$ is the oscillation period of a single particle subjected only to a harmonic trap with confinement frequency $\omega$. We will address the origin of this quasi-breathing.

## 1.2  Intuition from one-dimensional breathers in a scale-invariant gas

To develop some intuition, it is instructive to look at a similar phenomenon in a one-dimensional (1D) scale-invariant gas [2]. In this case, a gas with a chemical potential quadratic in density, $\mu \propto n^2$, with an initial rectangular density profile, was released to a harmonic trap. For the purposes of our paper, we will consider a situation where the gas is prepared in a state where the trapping energy does not evolve over time; that this is possible is guaranteed by scale invariance [3].

Under the above circumstances, the gas will breathe with a period $T_{\text{breathing}} = T/4$, where $T$ is the period of the harmonic trap. As in the case of triangular 2D breathers [1, 2, 4], the atomic cloud will be sharply divided onto the central "bulk" area and a shock-wave front. The former and the latter will be separated by a discontinuity line called the "inner shock-wave edge", itself located at $R_{\text{inner}}$; the shock-wave front will be separated from the outside vacuum by the "outer shock-wave edge", located at $R_{\text{outer}}$. The bulk density remains flat at all times, leading to a vanishing force; the velocity field is thus that of a free gas.

At the breathing half-period, i.e. at $t = T_{\text{breathing}}/2 = T/8$, the inner shock-wave edge reaches the origin: the two shock waves, one from the right edge and one from the left, collide. If hydrodynamic equations are used as an approximation, their solution ceases to exist at this point.

Here and below, $t^*$ will denote the instant of time when the inner edge reaches zero, regardless of whether periodic breathing exists or not. Note, however, that in one spatial dimension, the two times coincide:

$$\frac{T_{\text{breathing}}}{2} = t^*.$$

Also note that the initial state is time-reversal invariant. In this case, it is easy to show that the breathing half-period instant of time, $t = T_{\text{breathing}}/2$, also must be time-reversal invariant. An immediate consequence is that velocity field vanishes identically at this point.

Remarkably, the time-reversal invariance at $T_{\text{breathing}}/2$ also promotes the condition $T_{\text{breathing}}/2 = t^*$ from a curiosity to a necessity. This is because the velocity field of the bulk, $v_{\text{bulk}}(r, t)$, does not vanish as $t$ approaches $T_{\text{breathing}}/2$ from below, but instead approaches the value $-\omega r$. The only way for the time-reversal invariance to nevertheless hold at $T_{\text{breathing}}/2$ is if the bulk disappears at this point, meaning that $T_{\text{breathing}}/2$ coincides with $t^*$.

As a side remark, recall that the hydrodynamic solutions can not be continued beyond $t^*$. However, the hydrodynamic system can be considered an approximation to a more complete theory—either the Liouville equation for free fermions or the Gross-Pitaevskii equation with quintic non-linearity, both considered in Ref. [2]—and the hydrodynamics can be carried through the catastrophes. It would re-emerge from them with new initial conditions.

To summarize, for a 1D scale-invariant gas, the instant of time $t = T/8$ is, at the same time, a half of the breathing period, $T_{\text{breathing}}/2$ and a point $t^*$ where the inner edge of the shock wave front reaches the origin.

As we said, time-reversal invariance requires that the velocity field vanishes identically at $t = T_{\text{breathing}}/2 = t^*$. Let us list four trivial corollaries of the vanishing velocity field:

(A)      The bulk area, which has a nonzero velocity field, shrinks to a point;     (1)

(B)
$$\int d\text{Volume}\, n(r, t^*)\, v(r, t^*) \propto \langle v \rangle_{n(r, t^*)} = 0\,; \tag{2}$$

(C)
$$v(0, t^*) = 0\,; \tag{3}$$

(D)
$$v(R_{\text{outer}}(t^*), t^*) = 0\,; \tag{4}$$

where

$$\langle v \rangle_{n(r, t)} \equiv \frac{\int d\text{Volume}\, n(r, t)\, v(r, t)}{\int d\text{Volume}\, n(r, t)}$$

is a weighted average of the velocity field $v(r, t)$ with a non-negative weight distribution $n(r, t)/\int d\text{Volume}\, n(r, t)$. These four corollaries form a skeleton of our paper.

## 1.3   What to expect in higher spatial dimensions, for a scale-invariant gas

It is likely that, in higher spatial dimensions, scale-invariant gases do not exhibit breathing analogous to the period-$T/4$ breathing in 1D [2] (recall that the period-$2T$ breathing observed in Ref. [1] in 2D is a separate phenomenon). However, what *is* universal across different numbers of dimensions is the existence of an instant of time $t^*$ when the inner edge reaches the origin. We will show below that, while the velocity field does not vanish identically for $d > 1$ spatial dimensions, the properties (1-4) of Subsection 1.2 nevertheless continue to hold. As a result, the state of the system at $t = t^*$ emerges as an impostor of a breathing half-period, for any number of spatial dimensions, including the empirically relevant case $d = 2$.

## 1.4 The hydrodynamic equations

We will be working with the hydrodynamic equations (the continuity and Euler equations) that describe the dynamics of a $d$-dimensional gas at zero temperature:

$$
\begin{aligned}
\frac{\partial}{\partial t}n + \boldsymbol{\nabla}\cdot(n\boldsymbol{v}) &= 0\,, \\
\frac{\partial}{\partial t}\boldsymbol{v} + (\boldsymbol{v}\cdot\boldsymbol{\nabla})\boldsymbol{v} &= -\frac{1}{m}\boldsymbol{\nabla}\mu(n) - \omega^2\boldsymbol{r}\,,
\end{aligned}
\tag{5}
$$

where $n = n(\boldsymbol{r}, t)$ and $\boldsymbol{v} = \boldsymbol{v}(\boldsymbol{r}, t)$ are respectively the number density and the velocity fields, $\mu(n)$ is the chemical potential, $\omega$ is the frequency of the harmonic confinement, and $m$ is the particle mass.

We will be mainly interested in the spherically symmetric case:

$$
\begin{aligned}
\frac{\partial}{\partial t}n + \frac{1}{r^{d-1}}\frac{\partial}{\partial r}\left(r^{d-1}nv\right) &= 0\,, \\
\frac{\partial}{\partial t}v + v\frac{\partial}{\partial r}v &= -\frac{1}{m}\frac{\partial}{\partial r}(\mu(n)) - \omega^2 r\,.
\end{aligned}
\tag{6}
$$

Here we assume that the density is a function of the radial coordinate only, $n(\boldsymbol{r}, t) = n(r, t)$, and that the velocity field has a radial component only, itself exclusively a function of $r$: $\boldsymbol{v}(\boldsymbol{r}, t) = v(r, t)\boldsymbol{e}_r$, with $\boldsymbol{e}_r$ the unit vector in the radial direction. We will still occasionally address the more general system, as a tool in some of the proofs to follow.

# 2 Why the velocity field must vanish at $t = T_{\text{breathing}}/2$

Let us address in greater detail the question of time-reversal invariance at the breathing half-period—and its main consequence, the vanishing velocity field.

**Statement 1.** *If the system undergoes breathing of period $T_{\text{breathing}}$ when started from a time-reversal-invariant state at $t = 0$, then the velocity field vanishes identically at the breathing half-period:*

$$
v\left(r, \frac{T_{\text{breathing}}}{2}\right) = 0\,.
$$

*Proof.* We have that the time evolution is periodic with a period $T_{\text{breathing}}$,

$$
v(r, t + T_{\text{breathing}}) = v(r, t)\,.
$$

Also, the initial state is a time-reversal-invariant point, which means that

$$
v(r, -t) = -v(r, t)\,.
$$

In particular,

$$
v\left(r, +\frac{T_{\text{breathing}}}{2}\right) = -v\left(r, -\frac{T_{\text{breathing}}}{2}\right) = -v\left(r, +\frac{T_{\text{breathing}}}{2}\right)\,.
$$

It follows that

$$
v\left(r, +\frac{T_{\text{breathing}}}{2}\right) = 0\,,
$$

so the velocity field must vanish identically at $t = T_{\text{breathing}}/2$. $\square$

## 3 The shock wave and the bulk

We will be interested in the following initial conditions for the hydrodynamic equations (6):

$$n(r, 0) = \begin{cases} n_0 & \text{for } r \leq R_0, \\ 0 & \text{for } r > R_0, \end{cases} \tag{7}$$
$$v(r, 0) = 0.$$

That is, initially the gas cloud is a ball of radius $R_0$ and density $n_0$, at rest. Because the density has a discontinuity, it is an ill-posed problem to ask for a solution of the system in Eqs. (6) subject to the initial conditions in Eqs. (7). However, for a power-law equation of state,

$$\mu = a n^\nu, \tag{8}$$

there exists an approximate solution, valid at the initial stages of propagation—the so-called Damski-Chandrasekhar shock wave [4–6]—that converges to the initial condition (7) as $t \to 0^+$. This solution becomes exact if one replaces the $d$-dimensional divergence $r^{-(d-1)} \partial / \partial r (r^{(d-1)} \cdot)$ in the continuity equation (6) by the one-dimensinal derivative $\partial / \partial r$. It is not yet known if it is the *only* solution with this property. However, in a few particular cases, the Damski-Chandrasekhar solution is supported by a theory that is more regular than the hydrodynamic system in Eqs. (6), and of which Eqs. (6) are an approximation. The cases correspond to a scale-invariant gas, in one, two, and three spatial dimensions, with the underlying regular models given by a polynomial Gross-Pitaevskii equation, a free-fermionic Liouville's equation mappable to Eqs. (6), and a physical ultracold Bose gas [1,2]. Here and below, $a \geq 0$ is a constant. In our case, the Damski-Chandrasekhar solution can be obtained using the following initial condition at $t = \delta t > 0$, where $\delta t \to 0^+$ is a much shorter time interval than any time scale in the system:

$$
\begin{aligned}
n(r, \delta t) &\approx \begin{cases} n_0 & \text{for } r \leq R_0 + V_{\text{inner}}(0)\delta t \\ n_{\text{inter}}(r, \delta t) & \text{for } R_0 + V_{\text{inner}}(0)\delta t < r \leq R_0 + V_{\text{outer}}(0)\delta t \;, \\ 0 & \text{for } r > R_0 + V_{\text{outer}}(0)\delta t \end{cases} \\
v(r, \delta t) &\approx \begin{cases} 0 & \text{for } r \leq R_0 + V_{\text{inner}}(0)\delta t \\ v_{\text{inter}}(r, \delta t) & \text{for } R_0 + V_{\text{inner}}(0)\delta t < r \leq R_0 + V_{\text{outer}}(0)\delta t \;, \\ \text{undefined} & \text{for } r > R_0 + V_{\text{outer}}(0)\delta t \end{cases}
\end{aligned}
\tag{9}
$$

where

$$n_{\text{inter}}(r, \delta t) = \left[ \frac{m\nu}{a(\nu+2)^2} \left( \frac{r - (R_0 + V_{\text{outer}}(0)\delta t)}{\delta t} \right)^2 \right]^{\frac{1}{\nu}},$$
$$v_{\text{inter}}(r, \delta t) = \frac{2}{(\nu+2)} \left( \frac{r - (R_0 + V_{\text{outer}}(0)\delta t)}{\delta t} \right) + V_{\text{outer}}(0),$$

with

$$V_{\text{inner}}(0) = -c(n_0), \tag{10}$$

and

$$V_{\text{outer}}(0) = \frac{2}{\nu} c(n_0). \tag{11}$$

The velocities $V_{\text{inner}}(t) \equiv \dot{R}_{\text{inner}}(t)$ and $V_{\text{outer}}(t) \equiv \dot{R}_{\text{outer}}(t)$ correspond to the velocities of the inner and outer edges of the shockwave front, with positions respectively given by $R_{\text{inner}}(t)$ and $R_{\text{outer}}(t)$. The inner and outer edges are defined in Eq. (13), below. Here

$$c(n) = \sqrt{\frac{n}{m}\frac{\partial}{\partial n}\mu(n)} \overset{\mu \propto n^{\nu}}{=} \sqrt{\frac{\nu\mu(n)}{m}} \overset{\mu \propto n^{\frac{2}{d}}}{=} \sqrt{\frac{2\mu(n)}{d\,m}} \tag{12}$$

is the speed of sound. The state (9) does converge to the initial state (7) when $\delta t \to 0^+$. Also, when $d = 1$ and $\omega = 0$, the field (9) is, if $\delta t$ is replaced by $t$, an exact solution of the system (6). Again, for $d \neq 1$, the solution (9) becomes approximate, and it is the difference between the $d$-dimensional and one-dimensional divergences in the continuity equation, $r^{-1}(d-1)\cdot$, that makes it inexact. But since at the initial stages of the evolution, the density gradient and the velocity gradient diverge, the correction term can be neglected in comparison. On the other hand, Eq. (9) does not have discontinuities, and thus can be used as an initial condition for time propagation, for all times.

At later times, the initial condition (9) continues to evolve in such a way that at all instants of time before a certain critical time $t^*$, which we define later, there will be two distinct layers: the inner core (or, the "bulk") and an outer shell representing the "shock wave front." Note that under the harmonic confinement, the density in the bulk will remain spatially uniform at all times, and it will evolve as if the interactions did not exist. The hydrodynamic fields will then, for $0 < t \le t^*$, have the form

$$n(r, t) \approx \begin{cases} n_{\text{bulk}}(r, t) & \text{for } 0 \le r \le R_{\text{inner}}(t) \\ n_{\text{shockwave}}(r, t) & \text{for } R_{\text{inner}}(t) < r \le R_{\text{outer}}(t) \\ 0 & \text{for } r > R_{\text{outer}}(t) \end{cases},$$

$$\tag{13}$$

$$v(r, t) \approx \begin{cases} v_{\text{bulk}}(r, t) & \text{for } 0 \le r \le R_{\text{inner}}(t) \\ v_{\text{shockwave}}(r, t) & \text{for } R_{\text{inner}}(t) < r \le R_{\text{outer}}(t) \\ \text{undefined} & \text{for } r > R_{\text{outer}}(t) \end{cases},$$

with

$$\begin{aligned} n_{\text{bulk}}(r, t) &= \frac{n_0}{\cos^d(\omega t)} & \text{for } & r \le R_{\text{inner}}(t), \\ v_{\text{bulk}}(r, t) &= -\omega\tan(\omega t)r & \text{for } & r \le R_{\text{inner}}(t). \end{aligned} \tag{14}$$

While the shock wave from the density and velocity profiles $n_{\text{shockwave}}(r, t)$ and $v_{\text{shockwave}}(r, t)$ will require (for $d > 1$) a numerical treatment, in what follows we will be able to derive analytic expressions for the trajectories of the inner and outer edges of the shock wave front, $R_{\text{inner}}(t)$ and $R_{\text{outer}}(t)$.

## 4 The definition of $t^*$

We are particularly interested in the state of the system at the first instant when the inner edge reaches the origin and the bulk disappears; we denote this instant of time by $t^*$:

$$R_{\text{inner}}(t^*) = 0. \tag{15}$$

As we said in the introduction, the instant of time $t^*$ appears to be a good candidate for the half-period of an approximate breathing. In what follows, we will justify this assertion.

## 5   The property (A)

The property (A) from corollary (1) trivially follows from the definition of $t^*$ in Eq. (15):

**Statement 2** (Property (A)). *At $t = t^*$, the bulk region shrinks to a point.*

In spite of its triviality, the above property is important for establishing that the velocity field at $t^*$ is nearly zero. Indeed, according to (14), the bulk velocity field is not identically zero unless $t = (T/2) \times$ integer. However, the half-breathing can not happen at $T/2$, because the bulk density diverges prior to that, at $t = T/4$. The only remaining possibility is that the bulk region disappears altogether. And, according to the Statement above, this is precisely what happens at $t = t^*$.

Recall that in the one-dimensional case [2], where the true breathing is present, the above scenario is realized *verbatim*.

## 6   The property (B)

The property (B) of corollary (2) is specific to scale-invariant gases with a stationary moment of inertia.

For scale-invariant gases,

$$\nu = \frac{2}{d},$$

the dynamics of the moment of inertia,

$$J(t) \equiv m \int d\,\text{Volume}\, n(\boldsymbol{r}, t)\, r^2,$$

decouples from the dynamics of the rest of the system [3]. In particular, if the trapping frequency is adjusted in such a way that the energy of the gas before the trap was switched on equals the trapping energy right after,

$$\epsilon(n_0) = \frac{1}{2}\omega^2 \langle r^2 \rangle(0), \tag{16}$$

then the moment of inertia remains stationary at all times. Here and below,

$$\epsilon(n) = \frac{1}{n}\int_0^n dn' \mu(n') \overset{\mu \propto n^\nu}{=} \frac{1}{1+\nu}\mu(n) \overset{\mu \propto n^{\frac{2}{d}}}{=} \frac{d}{d+2}\mu(n) \tag{17}$$

is the energy per particle, and

$$\sqrt{\langle r^2 \rangle(t)} \equiv \sqrt{N^{-1}\int d\,\text{Volume}\, n(r, t)\, r^2}$$

is the r.m.s. distance to the origin. Here and below,

$$N \equiv \int d\,\text{Volume}\, n(r, t)$$

is the number of particles.

When the initial state is a ball of uniform density $n_0$ and radius $R_0$, the condition (16), in combination with (17), will fix the ball radius to

$$R_0 = \sqrt{2}\sqrt{\frac{\mu(n_0)}{m\omega^2}}. \tag{18}$$

The generator of the scaling transformations,

$$Q(t) \equiv m \int d\text{Volume}\, n(\boldsymbol{r}, t)(\boldsymbol{r} \cdot \boldsymbol{v}(r, t)),$$

is proportional to the time derivative of the moment of inertia [3]. In the state (16) where the moment of inertia is stationary, we have that $Q = 0$. This leads to

**Statement 3** (Property (B)). *At $t = t^*$, the following integral vanishes:*

$$\int dr\, r^d\, n(r, t^*) v(r, t^*) = 0.$$

## 7 A useful corollary of scale invariance

Scale invariance induces a similarity between the real and velocity spaces. A scale-invariant gas, characterized by a chemical potential

$$\mu(n) = an^{\frac{2}{d}},$$

($a$ being the coupling constant), has the same equation of state as free zero-temperature fermions. In the latter system, the chemical potential becomes the Fermi energy, the kinetic energy on the Fermi surface in the velocity space. The energy per particle becomes the mean velocity square. These observations inspire the following Statement:

**Statement 4.** *For a ball of* scale-invariant *gas of radius $R_0$ and uniform density $n_0$,*

$$\frac{\epsilon(n_0)}{\mu(n_0)} = \frac{\langle r^2 \rangle}{R_0^2} = \frac{d}{d+2}. \tag{19}$$

*Proof.* The relationship (19) can be proven using a direct computation. On one hand, according to (17), $\epsilon = \mu d/(d+2)$. On the other hand, one can directly verify that the mean square distance to the origin in a $d$-dimensional uniformly filled ball is

$$\langle r^2 \rangle = \frac{\int d\text{Volume}\, n_0 r^2}{\int d\text{Volume}\, n_0} = \frac{\int_0^{R_0} dr\, r^{d+1}}{\int_0^{R_0} dr\, r^{d-1}} = \frac{d}{d+2} R_0^2.$$

$\square$

## 8 The property (C)

It is also easy to verify the property (C) of corollary (3):

**Statement 5** (Property (C)). *At $t = t^*$, the velocity field vanishes at the origin:*

$$v(0, t^*) = 0.$$

*Proof.* In fact, at *any* instant of time, velocity at the origin is zero. This follows from the single-valuedness of the field $\mathbf{v}(\mathbf{r}, t)$ in Eqs. (5). Let us prove it using *reductio ad absurdum*. In the case of cylindrical symmetry, the velocity field has the form $\mathbf{v}(r, \phi) = v(r)\mathbf{e}_r(\phi)$, where $r \equiv \sqrt{x^2 + y^2}$ and $\phi = \arg(x, y)$ are the cylindrical coordinates, $\mathbf{e}_r(\phi) = \cos(\phi)\mathbf{e}_x + \sin(\phi)\mathbf{e}_y$ is the radial unit vector, and $\mathbf{e}_x$ and $\mathbf{e}_y$ are the unit vectors along the $x$ and $y$ directions, respectively. Assume that $v(0) \neq 0$. In that case, the $x$- and $y$-components of the velocity vector at the origin, $v_x(0, \phi) = v(0)\cos(\phi)$ and $v_y(0, \phi) = v(0)\sin(\phi)$, will depend on $\phi$. But the origin is a single point in space, and the values of the velocity components must be defined there unambiguously. Hence $v(0) = 0$. □

## 9 The inner edge dynamics: deriving an expression for $t^*$

Let us consider an auxiliary problem, where the trap is removed:

$$\frac{\partial}{\partial t}\bar{n} + \frac{\partial}{\partial r}(\bar{n}\bar{v}) + \frac{d-1}{r}\bar{n}\bar{v} = 0, \tag{20}$$

$$\frac{\partial}{\partial t}\bar{v} + \bar{v}\frac{\partial}{\partial r}\bar{v} = -\frac{1}{m}\frac{\partial}{\partial r}(\mu(\bar{n})). \tag{21}$$

We will focus on the scale-invariant case,

$$\mu(n) = an^{\frac{2}{d}}.$$

It follows from scale-invariance [3] that any solution of the auxiliary problem can be used to generate a solution of the actual problem [4]. In addition, we will assume that the initial state satisfies the condition of stationarity of the moment of inertia (16). We get

$$
\begin{aligned}
n(r, t) &= \frac{1}{\cos^d(\omega t)}\bar{n}(r/\cos(\omega t), \, \omega^{-1}\tan(\omega t)), \\
v(r, t) &= \frac{1}{\cos(\omega t)}\bar{v}(r/\cos(\omega t), \, \omega^{-1}\tan(\omega t)) - \omega r\tan(\omega t).
\end{aligned}
\tag{22}
$$

In particular, the trajectory of the inner edge $\bar{R}_{\text{inner}}(t)$ in the auxiliary problem (20)-(21), which is a point of discontinuity in both the density and the velocity gradients, can be used to generate the corresponding trajectory for the problem in Eqs. (6):

$$R_{\text{inner}}(t) = \cos(\omega t)\bar{R}_{\text{inner}}(\omega^{-1}\tan(\omega t)). \tag{23}$$

For the auxiliary problem (20)-(21), we will use the same initial condition (7) as for the actual problem (6). Remark that, in the auxiliary problem, the density in the bulk area will remain equal to $n_0$, and the velocity will remain equal to zero, at all times:

$$
\begin{aligned}
\bar{n}(r, t) &= n_0 \quad \text{for} \quad 0 \le r \le \bar{R}_{\text{inner}}(t), \\
\bar{v}(r, t) &= 0 \quad \text{for} \quad 0 \le r \le \bar{R}_{\text{inner}}(t).
\end{aligned}
\tag{24}
$$

Let us now focus on the zone to the right from $\bar{R}_{\text{inner}}(t)$. A Taylor expansion in the powers of $r - \bar{R}_{\text{inner}}(t)$ gives

$$
\begin{aligned}
\bar{n}(r, t) &= n_0 + \bar{n}_1(t)(r - \bar{R}_{\text{inner}}(t)) + \cdots, \\
\bar{v}(r, t) &= \bar{v}_1(r - \bar{R}_{\text{inner}}(t)) + \cdots, \\
\frac{1}{r} &= \frac{1}{\bar{R}_{\text{inner}}(t)} - \frac{r - \bar{R}_{\text{inner}}(t)}{\bar{R}_{\text{inner}}^2(t)} + \cdots, \qquad r > \bar{R}_{\text{inner}}(t).
\end{aligned}
$$

In addition, consider

$$\bar{\mu}(r, t) = \mu(n_0) + \bar{\mu}_1(t)(r - \bar{R}_{\text{inner}}(t)) + \cdots, \qquad r > \bar{R}_{\text{inner}}(t).$$

Equation (20), in the zeroth order in $(r - \bar{R}_{\text{inner}}(t))$, gives

$$\bar{n}_1 \dot{\bar{R}}_{\text{inner}} = n_0 \bar{v}_1. \tag{25}$$

Equation (21), in the zeroth order, leads to $\bar{v}_1 \dot{\bar{R}}_{\text{inner}} = \frac{1}{m} \bar{\mu}_1$ or, using (25),

$$m \dot{\bar{R}}_{\text{inner}}^2 = \frac{n_0}{\bar{n}_1} \bar{\mu}_1. \tag{26}$$

Now, according to (12),

$$\bar{\mu}_1 = \left. \frac{\partial \mu}{\partial n} \right|_{n=n_0} \bar{n}_1 = mc^2(n_0) \frac{\bar{n}_1}{n_0}. \tag{27}$$

Substituting (27) to (26), we get

$$\dot{\bar{R}}_{\text{inner}} = \pm c(n_0), \tag{28}$$

with the lower sign corresponding to the solution we are looking for. We get

$$\bar{R}_{\text{inner}}(t) = R_0 - c(n_0)t.$$

Curiously, in the no-trap case, the above result appears to be quite general, i.e. applicable to any equation of state $\mu(n)$.

Finally, with the help of the map (23), we arrive at the following:

**Statement 6.** *For* scale-invariant *gases in any number of spatial dimensions d, the inner edge moves as if it were a free particle:*

$$R_{\text{inner}}(t) = R_0 \cos(\omega t) + \omega^{-1} V_{\text{inner}}(0) \sin(\omega t), \tag{29}$$

*with the initial velocity given by*

$$V_{\text{inner}}(0) = -c(n_0).$$

Note that the value for the initial velocity of the inner edge is consistent with the one in (10): the latter has been obtained independently, using the Damski map [4,6] to the Chandrasekhar solution [5] of the nonlinear transport equation.

Let us now define

$$t^* = \frac{1}{\omega} \arctan\left(\frac{\omega R_0}{|V_{\text{inner}}(0)|}\right) = \frac{\arctan(\sqrt{d})}{2\pi} T \tag{30}$$

as the instant when the inner edge reaches the origin for the first time. In the second line above, we used (18) and (12). We assume $V_{\text{inner}}(0) < 0$.

In Fig. 1, we compare the prediction (29) with the results of a numerical propagation of the hydrodynamic equations (6), in the two-dimensional case.

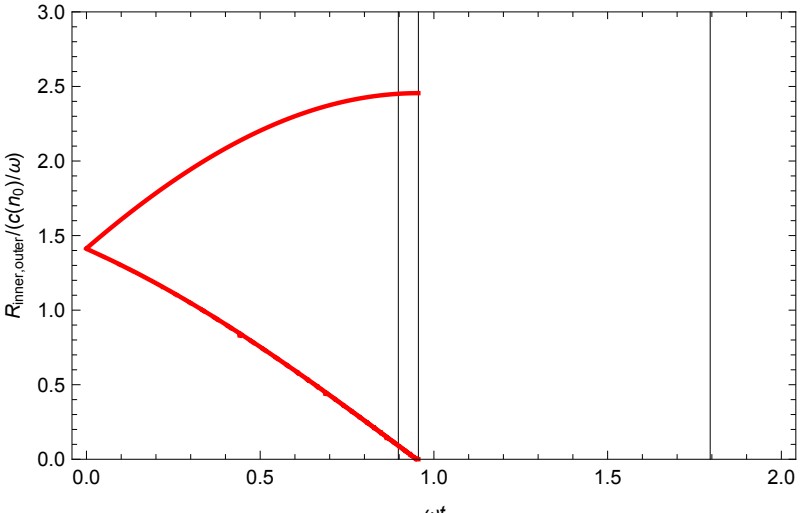

Figure 1: Numerically computed trajectories of the outer (upper curve) and inner (lower curve) edges of the shock wave front, together with the analytical predictions (29) and (35). The numerical curves are indistinct from the analytical predictions. The three thin vertical lines correspond to the following instants of time, in order from left to right: the experimentally observed half quasi-breathing period $T/7$ (see [1]), the hydrodynamic prediction for a half quasi-breathing period (30), and the full experimental quasi-breathing period $2T/7$ observed in [1]. As we show below (see e.g. Fig. 2), at the instant of time $t^*$ from (30), the density distribution develops a conical singularity at the origin, and so the hydrodynamical equations can not be propagated past this moment in time.

## 10 The outer edge dynamics: an expression for $t^*_{\text{outer}}$

Let us start with a particular reformulation of the hydrodynamic set of equations (6), specific for power-law equations of state, $\mu(n) \propto n^\nu$:

$$\frac{\partial}{\partial t}\mu + \left(\frac{\partial}{\partial r}\mu\right)v + \nu\mu\left(\frac{\partial}{\partial r}v\right) + \frac{\nu(d-1)}{r}\mu v = 0, \tag{31}$$

$$\frac{\partial}{\partial t}v + v\frac{\partial}{\partial r}v = -\frac{1}{m}\left(\frac{\partial}{\partial r}\mu\right) - \omega^2 r. \tag{32}$$

Assume that the motion of the outer shock-wave front edge—the point where the density vanishes—follows the trajectory $R_{\text{outer}}(t)$. Consider a Taylor expansion of both $\mu$ and $v$ fields in the powers of the distance to the outer edge:

$$\mu(r, t) = \tilde{\mu}_1(t)(r - R_{\text{outer}}(t)) + \tilde{\mu}_2(r - R_{\text{outer}}(t))^2 + \cdots,$$
$$v(r, t) = \tilde{v}_0(t) + \tilde{v}_1(r - R_{\text{outer}}(t)) + \cdots,$$
$$\frac{1}{r} = \frac{1}{R_{\text{outer}}(t)} - \frac{r - R_{\text{outer}}(t)}{R_{\text{outer}}^2(t)} + \cdots.$$

Our goal is to prove that if $\tilde{\mu}_1(0) = 0$, then $\tilde{\mu}_1(t) = 0$ for all subsequent times. Once proven, this will mean that the force acting on a probe particle at the outer edge vanishes identically, and the outer edge will move as if it were a free particle.

From (32) we get, to the zeroth order,

$$\dot{v}_0 = -\frac{1}{m}\tilde{\mu}_1 - \omega^2 R_{\text{outer}}.$$

Equation (31), in the zero order, yields

$$\tilde{v}_0 = \dot{R}_{\text{outer}},$$

and hence

$$\ddot{R}_{\text{outer}} = -\frac{1}{m}\tilde{\mu}_1 - \omega^2 R_{\text{outer}}^2. \tag{33}$$

In the first order, Eq. (31) gives

$$\dot{\tilde{\mu}}_1 + (1+\nu)\tilde{v}_1\tilde{\mu}_1 - \frac{\nu(d-1)}{R_{\text{outer}}}\dot{R}_{\text{outer}}\tilde{\mu}_1 = 0.$$

The equation above is a linear homogeneous time-dependent ordinary differential equation for $\tilde{\mu}_1(t)$. Given the initial state of the system (9), the initial condition is $\tilde{\mu}_1(0) = 0$. Therefore, $\tilde{\mu}_1$ remains identically zero at all times:

$$\tilde{\mu}_1(t) = 0.$$

Equation (33) then becomes Newton's equation for a free particle:

$$\ddot{R}_{\text{outer}} = -\omega^2 R_{\text{outer}}^2. \tag{34}$$

This brings us to the following statement:

**Statement 7.** *For any power-law equation of state $\mu(n) \propto n^\nu$, in any number of spatial dimensions $d$,*

$$R_{\text{outer}}(t) = R_0 \cos(\omega t) + \omega^{-1} V_{\text{outer}}(0) \sin(\omega t). \tag{35}$$

For future use, let us further define

$$t_{\text{outer}}^* = \frac{1}{\omega} \arctan\left(\frac{V_{\text{outer}}(0)}{\omega R_0}\right) \tag{36}$$

as the point in time when the outer edge stops for the first time. We will be assuming that $V_{\text{inner}}(0) > 0$.

## 11 The property (D)

Unexpectedly, the property (D) of corollary (4) also turns out to be satisfied, and it seems to be a pure coincidence:

**Statement 8** (Property (D)). *For any scale-invariant equation of state $\mu(n) \propto n^{\frac{2}{d}}$, in any number of spatial dimensions $d$,*

$$V_{\text{outer}}(t^*) = 0.$$

*Equivalently, under these conditions,*

$$t_{\text{outer}}^* = t^*, \tag{37}$$

*or, from (30) and (36),*

$$\frac{|V_{\text{inner}}(0)|V_{\text{outer}}(0)}{\omega^2 R_0^2} = 1. \tag{38}$$

*Proof.* We can prove the Eq. (37) using a combination of (10), (11), (19), and (16). □

Note that at $t = t^*$, the outer radius reaches

$$R_{\text{outer}}(t^*) = \sqrt{d+1}\, R_0,$$

for any number of spatial dimensions $d$.

## 12 Summary and numerical results

We showed that when a uniformly filled ball of a $d$-dimensional scale-invariant gas is released to a harmonic trap of period $T$, the state of the gas at

$$t^* = \left( \frac{\arctan \sqrt{d}}{2\pi} \right) T$$

emerges as a 'skillful impostor' of a state that *would* emerge at a half of a breathing period, if a breathing of period

$$T_{\text{quasi-breathing}} = 2\, t^* = \left( \frac{\arctan \sqrt{d}}{\pi} \right) T$$

were present in the system.

More specifically, we show that at $t = t^*$, the velocity field almost vanishes , because it is constrained by two zeros and a vanishing weighted average; meanwile, the bulk area (whose velocity is never zero) shrinks to a point. Here the relevant velocity scale to compare the velocity field with is the initial speed of sound, $c(n_0)$.

While the hydrodynamics itself breaks down at $t = t^*$, a vanishing velocity field could allow a more general theory or a physical system—Liouville equation for the underlying free fermions [2], a Gross-Pitaevskii equation [1,2], or an ultracold quantum gas [1]—to continue to evolve in time in a manner that both (i) supports the time-reversal invariance and (ii) does not lead to temporal discontinuities in the velocity field.

The breathing of period $(\arctan(\sqrt{d})/\pi)\, T$ is an exact phenomenon in the one-dimensional case [2]. In two spatial dimensions, a quasi-breathing of (quasi-) period $2T/7 = 0.286\ldots \times T \approx (\arctan(\sqrt{2})/\pi)\, T = 0.304\ldots \times T$ has been observed both experimentally and numerically in Ref. [1].

We have confirmed our analytical predictions by numerical results. In two spatial dimensions, at the moment $t^*$, which is associated with the quasi-breathing half-period $T_{\text{quasi-breathing}}$, the central "bulk" area shrinks to a point (Fig. 2), and the velocity uniformly falls significantly below the speed of sound, which is the relevant velocity scale (Fig. 3).

In addition, in Fig. 2, we test our hydrodynamical model (5) against the more empirically accurate Gross-Pitaevskii equation:

$$i\hbar \frac{\partial}{\partial t}\psi = -\frac{\hbar^2}{2m}\Delta\psi + g|\psi|^2\psi + \frac{m\omega^2}{2}r^2\psi\,,$$

$$\int |\psi|^2\, d^2\boldsymbol{r} = N\,, \qquad \psi = \psi(\boldsymbol{r}, t)\,.$$

Here and below, $m$ is the atomic mass, $g$ is the Gross-Pitaevskii coupling constant, $N$ is the number of atoms, and $\omega$ is the trapping frequency. The coupling constant was set to $g = a$, where $a$ is the coupling constant entering the equation of state (8). Recall that the Gross-Pitaevskii case corresponds to (8) with $\nu = 1$. The initial shape of the cloud was the ground state of a flat-bottom circular corral of a radius $R_0$. At $t = 0^+$, the corral walls were removed and the harmonic trap was turned on. The healing length $\xi \equiv \hbar/\sqrt{mg|\psi(\boldsymbol{0}, 0)|^2}$ was chosen to be $\xi = 0.016 R_0$, close the conditions of the experiment [1]. Similarly to the hydrodynamical case, the total number of particles was adjusted to a value for which the initial interaction energy was equal to the initial trapping energy.

Apart from the purely quantum-mechanical fringes, the agreement is quite satisfactory.

A similar comparison would be harder for the velocity plot Fig. 3: quantum-mechanically, velocity at a given spatial point is ill-defined.

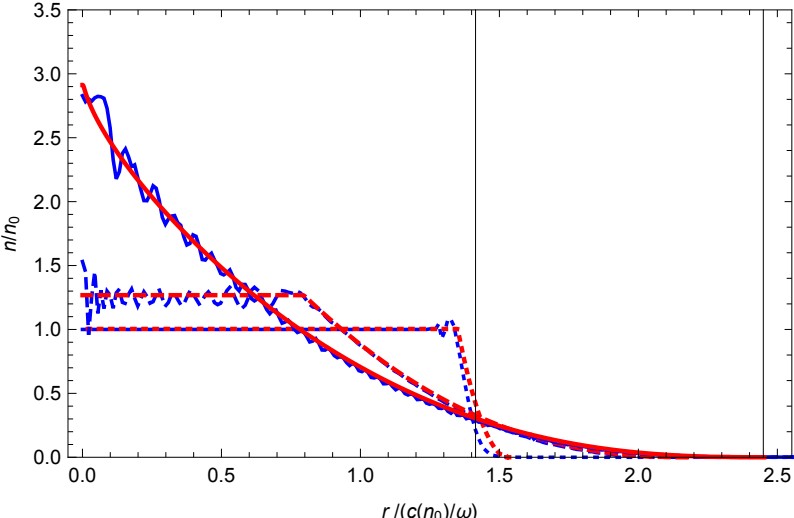

Figure 2: A two-dimensional disk-shaped quasi-breather. Shown are the density distributions at $\omega t = 0.0615924$ (short-dashed), $\omega t = 0.478$ (long-dashed), and $\omega t = \arctan(\sqrt{2}) = \omega T_{\text{quasi-breathing}}/2 \approx 0.955$ (solid). Recall that the quasi-revival period conjectured in Ref. [1] is $T'_{\text{quasi-breathing}} = 2T/7$, giving $\omega T'_{\text{quasi-breathing}}/2 = 0.898$. Both the hydrodynamic results (red) and the ab initio Gross-Pitaevskii calculation (blue) are shown. The two thin vertical lines correspond to, from left to right, the initial ball radius and the position of the outer edge of the atomic cloud at the hydrodynamic prediction for the quasi-breathing half-period (30).

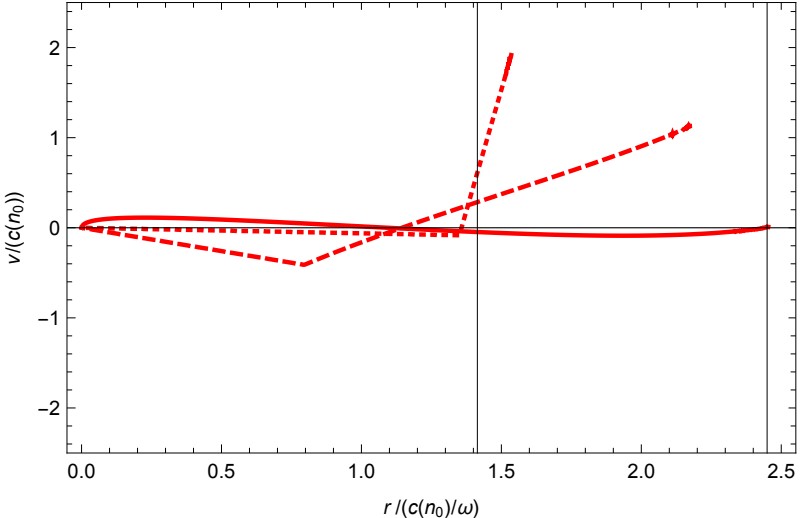

Figure 3: The velocity field as a function of the spatial coordinate, in a two-dimensional disk-shaped quasi-breather. The observation times, the line dashing and color convention, and the meaning of the thin vertical lines are the same as in Fig. 2.

## 13 Outlook

Let us re-analyze the properties of the state of the system at $t = t^*$, i.e. at the instant when the inner edge of the shock wave front touches zero (see Subsection 1.2). The vanishing bulk area is a trivial consequence of the definition of $t^*$. We discussed several zeros of the velocity field. The first zero, at $t = t^*$, is the trivial zero at the origin, which is required by the single-valuedness of the velocity field. The second zero is the vanishing of weighted average, which—up to a

constant—is nothing else but the well-known generator of the scaling transformations. This vanishes for the initial conditions that we have chosen [3]. Finally, there is a second zero of the velocity field, at the outer rim of the cloud, and most of our paper is devoted to it. But while we *do* prove that a zero emerges there—for all numbers of spatial dimensions—we still do not understand *why* this happens: the equality (37) looks like a pure coincidence.

A potential way to promote Eq. (37) from a coincidence to a necessity may look as follows:

1. For a scale-invariant gas in one dimension: $\mu(n) \propto n^2$, and the relationship (37) follows form the Shi-Gao-Zhai Fermi-Bose correspondence [2];

2. One may try to find a fundamental principle according to which for any power-law equation of state $\mu \propto n^{\nu}$, the product $|V_{\text{inner}}(0)|V_{\text{outer}}(0)$ does not depend on $\nu$;

3. A combination of the items 1 and 2 above proves (37), through (38).

# Acknowledgements

This work would not be possible without numerous discussions with Jean Dalibard, Zhe-Yu Shi, and Bogdan Damski.

**Funding information** This work was supported by the NSF (Grants No. PHY-1912542, and No. PHY-1607221) and the Binational (U.S.-Israel) Science Foundation (Grant No. 2015616). J.T.'s project PID2019-106290-C22 is financed by Ministerio de Ciencia e Innovación de España. M.G. is partially supported by the Spanish grant PGC2018-098676-B-I00 (AEI/FEDER/UE) and the Juan de la Cierva-Incorporación fellowship IJCI-2016-29071. G.E.A. acknowledges financial support from the Spanish MINECO (FIS2017-84114-C2-1-P), and from the Secretaria d'Universitats i Recerca del Departament d'Empresa i Coneixement de la Generalitat de Catalunya within the ERDF Operational Program of Catalunya (project QuantumCat, Ref. 001-P-001644).

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
