# Peer review of "The origin of the period-$2T/7$ quasi-breathing in disk-shaped Gross-Pitaevskii breathers"

_SciPost Physics, doi:SciPost Phys. 12, 092 (2022)_

## Round 1 · Referee Report · Anonymous (Referee 1) · 2021-9-27

Report

The manuscript by Torrents et al. addresses the origin of the 2T/7 quasi-period of the disk-shaped breather that was observed in trapped two-dimensional BEC by Dalibard’s group. In this paper, the authors have brilliantly argued that at a certain time t*, the velocity field almost vanishes, which naturally leads to a 2T/7 quasi-period. The key step of their argument is to focus on the trajectories of the singularities of the density and velocity fields (R_{inner} and R_{outer}) instead of the whole solution to the hydrodynamic equations. By doing this, they are able to extract the exact trajectories of both R_{inner} and R_{outer} and hence prove that the velocity field at these two points vanishes at t*. Since this work solves an experimental puzzle and the method is quite innovative, I think this paper can be published in Scipost.

Requested changes

Minor comments:
1.In eq.(5), a $\mathbf{v}$ is missed on the L.H.S. of the Euler’s equation.
2.Last paragraph on page 12, 2/7 should be around 2.86, instead of 2.96.
3.The converse of statement 1 (If the velocity field vanishes at T/2, then T is a period of the breather.) should also be true. Even though it is not directly related to the following argument in the paper, it will be nice that the authors add it to statement 1 to make things complete.

  • validity: top
  • significance: top
  • originality: top
  • clarity: top
  • formatting: excellent
  • grammar: perfect

Author:  Vanja Dunjko  on 2021-11-16  [id 1946]

(in reply to Report 1 on 2021-09-27)

We thank the referee for this very encouraging review: our replies are below.

Requested changes:

  1. Corrected.
  2. Corrected.
  3. We do not think this is correct in general. At the very least, we do not see a fundamental reason why there should not be more than two time-reversal-invariant instants of time per period. Incidentally, for the triangular breathers of [Phys. Rev. X vol. 9, 021035 (2019)], the instants of time $T_{\text{breathing}}/4$ and $3T_{\text{breathing}} /4$ come pretty close, exhibiting an identically vanishing velocity field along six separate rays; the rays are aligned with the corners of the right pentagon formed by the bulk density region at these moments. For this breather, $T_{\text{breathing}} = T /2$.

---

## Round 1 · Referee Report · Anonymous (Referee 2) · 2021-10-8

Strengths

1- The manuscript provides an elegant theoretical explanation of some aspects of the important experiment of the Jean Dalibard group.

2- The analytical approach employed for the study of the dynamics of an initially discontinuous density profile is highly original.

3- The manuscript provides intriguing results that should inspire further studies.

Weaknesses

1- The manuscript does not discuss differences between the numerical Gross-Pitaevskii solutions and the hydrodynamic solutions.

Report

The manuscript theoretically addresses an important problem posed by the puzzling experimental work of the Paris group. It (analytically) delivers a quite general, sort of surprising, result for the breathing "half-period", which in two spatial dimensions, surprisingly well, reproduces the experimental measurement. It is rather doubtful that such a result could be obtained with less sophisticated analytical techniques. I think that it definitely deserves publication in a high-impact journal such as SciPost after consideration of the remarks from the Requested Changes sections.

Requested changes

1-The left-hand side of (2) does not match the expression proportional to $C$. I suppose the definition of $C$ needs to be corrected. The brackets, ${\cdots}$, look awkward in (2), etc.

2-There is velocity field missing in one place in (5).

3-If (8) satisfies (6), after setting $\omega=0$ and $\delta t=t$, only when $d=1$ , then how is (8) found for an arbitrary $d$? Where does it come from? What fixes its form?

4- Below (11): $0+\to0^+$.

5- Should there be "$\text{for} \ r>R_\text{outer}(t)$" in one of the expressions in (12)?

6- What does the statement "the velocity field at $t^*$ is nearly zero" mean (remark right below Statement 2)? In what sense it is nearly zero?

7- I am not sure whether $N$, in the equation for $\sqrt{\langle r^2\rangle(t)}$, is somewhere defined.

8- Could it be explained why single-valuedness of the velocity field leads to the proof of Statement 5?

9- Should there be $\bar{R}_\text{inner}(t)$ instead of $R_\text{inner}(t)$ in the expression for $1/r$, the one below (23)?

10- It is unclear to me why the empty space, the $\omega t\gtrsim1$ region, is displayed in Fig. 1. Can analytical solutions be put on the plot as, e.g., the dashed line? Change "(34)-(28)" in the caption to, e.g., "(28,34)".

11- Should it be said in Statement 7 that (34) holds when $\mu(n)\sim n^\nu$? Similar question applies to Statement 8.

12- Should one explicitely define $V_\text{outer}(t)$ as $dR_\text{outer}(t)/dt$?

13- Figs. 2 and 3: the "dotted curves" seem to me to be rather "dashed" than "dotted".

14- Fig. 3: why the blue, Gross-Pitaevskii results, are missing? Smaller range of the vertical axis could lead to better presentation of the results.

15- Can Gross-Pitaevskii equation be written down? Can one comment on how its simulations are carried out (e.g. write explicitly what initial conditions for the numerics are employed)? Are there some specific parameters used for the Gross-Pitaevskii-based simulations?

  • validity: high
  • significance: high
  • originality: top
  • clarity: high
  • formatting: good
  • grammar: excellent

Author:  Vanja Dunjko  on 2021-11-16  [id 1945]

(in reply to Report 2 on 2021-10-08)

We wholeheartedly thank the referee for the appreciation of our work and for the extremely detailed reading and analysis. Below, we describe the way we are going to incorporate the suggestions that were offered.

Requested changes

  1. Corrected.

  2. Thank you, corrected.

  3. We have added an explanation, after Eq. 12.

  4. Corrected.

  5. Corrected.

  6. We compare the velocity at \(t^{*}\) with the typical velocity inside the shockwave front, in the middle of the time evolution. The relevant velocity scale is the initial speed of sound. We added an explanation, in the second paragraph of the summary.

  7. Corrected.

  8. We substantially lengthened the proof.

  9. Your help is truly invaluable. Corrected.

  10. This is an important point:

    1. We added an explanation to the caption of Fig. 1.

    2. In short, the hydrodynamical solution ceases to exist at \(t^{*}\) . The reason is a conical singularity emerging at the origin, at this instant of time.

    3. If the velocity were vanishing exactly, one might use the time-reversal invariants to carry the hydrodynamics through the catastrophe. But since the velocity vanishing is not exact, one needs a theory one level higher to restart hydrodynamics (with new initial conditions) after such a “Big Crunch/Big Bang”. The Gross-Pitaevslii equation does precisely that.

    4. As the ENS experiment and numerics show, the time evolution after \(t^{*}\) is both regular and erratic: there are regular quasi-revivals at integer multiples of \(2T /7\), but also there are quasi-random small modifications of the initial state at the quasi-revival points. Nonetheless, at \(t = 7(2T /7) = 2T\) , the revival is exact.

    A small remark: formally speaking, the upper curve (outer edge as a function of time) in Fig. 1 could be continued for a bit further, as the singularity at the origin will need some time to propagate to the edges. But we believe that such a continuation would be misleading.

  11. This was a significant omission on our part. Corrected in both places, thank you.

  12. Introduced, between the Eqs. 11 and 12.

  13. Corrected.

  14. We decided against attempting to introduce a quantum-mechanical analogue of the local velocity. We added a sentence stating this, at the end of the “Summary and numerical results” section.

  15. We added a paragraph at the end of the “Summary and numerical results” section. For this comment, we are especially thankful: it reminded us to mention, in the paper, that our Gross-Pitaevskii numerics aimed to replicate the ENS experiment [Phys. Rev. X vol. 9, 021035 (2019)] verbatim.

---

## Round 2 · Referee Report · Anonymous (Referee 2) · 2021-11-21

Report

It is my impression that the points raised in my first report have been adequately taken into account in the revised manuscript. I have no further "urgent" comments.
  • validity: -
  • significance: -
  • originality: -
  • clarity: -
  • formatting: -
  • grammar: -

Author:  Maxim Olshanii  on 2022-01-03  [id 2063]

(in reply to Report 1 on 2021-11-21)
Category:
remark

Without all the hard work by the Referee 1, this article would not happen: thank you.

---

## Round 2 · Referee Report · Anonymous (Referee 1) · 2021-12-12

Report

I am satisfied with the revision and support for publication.
  • validity: -
  • significance: -
  • originality: -
  • clarity: -
  • formatting: -
  • grammar: -

Author:  Maxim Olshanii  on 2022-01-03  [id 2062]

(in reply to Report 2 on 2021-12-12)
Category:
remark

We are grateful to the Referee 2 for all the help with the manuscript.

---

## Round 2 · List of Changes

1. Fixed Eq. (2) to address Referee 2's requested change 1, namely, we removed the proportionality constant $C$ and instead used the proportionality symbol $\propto$. Also, we removed the braces.

  2. Corrected a typo in the second equation in Eq. (5), namely the missing velocity field after the del operator on the left-hand side. This was Referee 1's requested change 1 and Referee 2's requested change 2.

  3. The power-law equation of state is now numbered as Eq. (8).

  4. After Eq. 8 (formerly 7), we clarified that the Damski-Chandrasekhar shock wave solution is approximate and valid at the initial stages of propagation.

  5. Right before Eq. (12) (formerly 11), we explained that $V_{\text{inner}}(t) \equiv \dot{R}_{\text{inner}}(t)$ and $V_{\text{outer}}(t) \equiv \dot{R}_{\text{outer}}(t)$. This addresses Referee 2's requested change 12.

  6. Right after Eq. (12) (formerly 11), fixed the notation of the limit of delta t, as reqested by Referee 2's requested change 4.

  7. After Eq. (12), we added an explanation of why Eq. (9) (formerly 8) is an approximate solution of Eq. (6) for d>1, as reqested by Referee 2's requested change 3.

  8. In Eq. (12) (formerly 12), in the last equation, corrected 'for $R_{\text{outer}}$' to read 'for $r>R_{\text{outer}}$'. This was Referee 2's requested change 5.

  9. After Eq. (17) (formerly 16), we added an explanation that $N$, the number of particles, is a volume integral over $n(r,t)$. This addresses Referee 2's requested change 7.

  10. We added a substantial amount of detail to the proof of Statemet 5, that the single-valuedness of the velocity field implies that the velocity at the origin is zero. This addresses Referee 2's requested change 8.

  11. In the expression for $1/r$, below Eq. (24) (formerly 23), we replaced all appeances of $R_{\text{inner}}(t)$ by $\bar{R}_{\text{inner}}(t)$. This addresses Referee 2's requested change 9.

  12. In the caption to to Fig. 1, we corrected the way we refer to Eqs. (29) and (35) (formerly 28 and 34). This addresses one part of Referee 2's requested change 10.

  13. At the end of the caption to Fig. 1, we added an explanation of why the hydrodynamical equations cannot be propagated past $t^{∗}$. This addresses the other part of Referee 2's requested change 10.

  14. In the text of Statement 7, we clarified that the statement holds for power-law equations of state. This addresses one part of Referee 2's requested change 11.

  15. In the text of Statement 8, we also clarified that the statement holds for power-law equations of state. This addresses the other part of Referee 2's requested change 11.

  16. After the second displayed equation in Sec. 12, we added a statement explaining that the relevant velocity scale to compare the velocity field with is the initial speed of sound. This addresses Referee 2's requested change 6.

  17. Near the end of the page where Sec. 12 begins, we corrected $2T/7 = 0.296\ldots \times T$ to $2T/7 = 0.286\ldots \times T$. This addresses Referee 1's requested change 2.

  18. In the caption of Fig. 2, we changed the description of lines from 'dotted' and 'dashed' to 'short-dashed' and 'long-dashed'. This addresses Referee 2's requested change 13.

  19. At the end of Sec. 12, we wrote down the Gross-Pitaevskii equation, with an explanation of the relevant parameters. We emphasized that the healing length and the number of particles were chosen to be close to the values they had in the ENS experiment [Phys. Rev. X vol. 9, 021035 (2019)]. This addresses Referee 2's requested change 15.

  20. At the end of Sec. 12, we added a sentence explaining why we decided against attempting to introduce a quantum-mechanical analogue of the local velocity. This addresses Referee 2's requested change 14.

---

## Editorial Decision

published